# Influence of Repair Welding on the Fatigue Behavior of S355J2 T-Joints

**DOI:** 10.3390/ma16103682

**Published:** 2023-05-11

**Authors:** Peng Zhao, Banglong Yu, Ping Wang, Yong Liu, Xiaoguo Song

**Affiliations:** 1School of Ocean Engineering, Harbin Institute of Technology, Weihai 264209, China; 2State Key Laboratory of Advanced Welding and Joining, Harbin Institute of Technology, Harbin 150001, China; 3Shandong Institute of Shipbuilding Technology, Weihai 264209, China

**Keywords:** orthotropic deck, repair welding, high cycle fatigue, traction structural stress

## Abstract

This paper investigated the effect of repair welding on the microstructure, mechanical properties, and high cycle fatigue properties of S355J2 steel T-joints in orthotropic bridge decks. The test results found that the increase in grain size of the coarse, heat-affected zone decreased the hardness of the welded joint by about 30 HV. The tensile strength of the repair-welded joints was reduced by 20 MPa compared to the welded joints. For the high cycle fatigue behavior, the fatigue life of repair-welded joints is lower than that of the welded joints under the same dynamic load. The fracture positions of toe repair-welded joints were all at the weld root, while the fracture positions of the deck repair-welded joints were at the weld toe and weld root, with the same proportion. The fatigue life of toe repair-welded joints is reduced more than that of deck repair-welded joints. The traction structural stress method was used to analyze fatigue data of the welded and repair-welded joints, and the influence of angular misalignment on was considered. The fatigue data with and without AM are all within the ±95% confidence interval of the master S-N curve.

## 1. Introduction

Orthotropic steel deck (OSD) is widely used in the construction of various bridges with the advantages of lightweight, low cost, and large bearing capacity [1,2]. However, fatigue cracks often appear on OSD due to welding residual stress and complex traffic load [3,4]. The cracks of the T-joints between the U-rib and deck have the greatest impact on OSD performance. Xiao et al. [5] conducted stress analysis and fatigue evaluation to understand key factors contributing to the cracks of rib-deck T-joints. The test results found that the fatigue cracks were mainly located at the weld toe and weld root. These fatigue cracks can be categorized into four types according to different growth paths, as illustrated in Figure 1, namely toe-deck crack (crack Ⅰ), root-deck crack (crack Ⅱ), root-weld crack (crack Ⅲ) and toe-rib crack (crack Ⅳ) [5]. Cracks I and II can penetrate the bridge panel and cause pavement damage [6,7]. Sim et al. [8] conducted a full-scale fatigue test on OSD. The test results found that the fatigue cracking of rib-deck T-joints was more serious at the deck side than at the rib side, and the cracking probability of the weld toe was higher than that of the weld root. Therefore, the research should focus on the fatigue cracks at the deck side.

At present, most of the bridges in service worldwide have used OSD structures [9], which are easily affected by fatigue cracks. In order to prolong the service life of the bridge, it is necessary to repair the fatigue crack in time. Among the many ways to repair cracks, repair welding is the most widely used with the advantages of good economy and high efficiency [10,11]. Numerous studies have been carried out to explore the influence of repair welding on the microstructure and mechanical properties of materials. Aman et al. [12] discovered a decrease in the hardness of the heat-affected zone (HAZ) after repair welding on 316 L stainless steel. The decrease in hardness was attributed to the reduction of δ-ferrite in HAZ, and the hardness will decrease with the increase of the number of repair welding. Luo et al. [13] investigated the effect of repair welding heat input on the microstructure of 304 stainless steel. It was found that with the increase of heat input, the δ-ferrite in the weld metal decreased slightly, while the austenite grain size in the HAZ was slightly increased. Katsas et al. [14] conducted tensile and bending tests on both welded joints and repair-welded joints. The results found that a single repair welding had no effect on the ultimate tensile strength of the welded joints. And in the absence of “non-fusion”, the repair-welded joints passed the bending test as successfully as the welded joints. Shojaati et al. [15] pointed out that multiple repair welding had no adverse effect on the microstructure and mechanical properties of X20Cr13 stainless steel. It can refine austenite grains in the HAZ and increase impact toughness. Dong [16] proposed that a major detrimental effect of repair welding on structural integrity is the increase of crack driving force caused by the elevation of membrane stress level, and the adjustment of welding size can effectively reduce the membrane stress.

Contrary to the research on microstructure and mechanical properties, there are few studies on the fatigue properties of repair-welded joints. Wang et al. [17] found that the fatigue life of the repair-welded specimen is 7.1 times that of the specimen using the drilling stop-hole method, and 3.3 times that of the specimen reinforced by carbon fiber-reinforced polymer (CFRP). Seo et al. [18] studied the fatigue property of the bogie structure after repair welding. The results found that the fatigue life of repair-welded joints was lower than that of welded joints. When the repair welding width decreased, the fatigue life increased.

Due to the structural complexity, current fatigue design/evaluation guidelines for welded structures are mostly stress range based, likely hot spot stress method [19,20] and nominal stress method [21,22]. A traction-based structural stress definition was introduced by imposing equilibrium conditions [23]. In addition to its demonstrated mesh-insensitivity and convenience when applied to large and complex assemblies, the method has led to the development of the Master S–N curve adopted by ASME Div 2 [24] since 2007.

In this study, the repair welding was carried out on S355J2 steel T-joints, and positions were selected at the weld toe and deck to fit actual working conditions. The microstructure and mechanical properties of the repair-welded joints were studied. In addition, the high cycle fatigue (HCF) properties of the joints are analyzed by the traction structural stress method considering the influence of angular misalignment (AM).

## 2. Experimental Procedure

### 2.1. Materials and Test Specimen Preparation

The material of base metal and filler metal used in this study are S355J2 steel and E500T-1 wire, respectively. Table 1 provides their chemical compositions. Metal active gas (MAG) arc welding is adopted in this study with the advantages of convenience on construction.

The manufacturing process and dimensions of test specimens are shown in Table 2. The size of the deck plate used in this study is 1800 mm × 600 mm × 18 mm (length × width × thickness), and the thickness of the U-rib is 8 mm. The blunt edge of the welding groove is 0–1 mm, and the groove Angle is 50°. The length of the weld is equal to the length of the deck plate, and the size of the weld leg is 8–10 mm. The OSD is cut by wire-cutting equipment to fabricate the T-joints. As shown in Table 2, the T-joint has a length of 300 mm and a width of 100 mm.

The welded joints, toe repair-welded joints, and root repair-welded joints are prepared in this study. The length of the repair weld is the same as the width of the deck plate. The width and depth of repair welds are 8 mm and 4.5 mm, respectively. Figure 2 shows the schematic diagrams of repair-welded joints.

### 2.2. Experiments

#### 2.2.1. Microstructure and Mechanical Properties Test

The T-joints were cut into metallographic samples by using a wire-cutting machine. The metallographic samples are corroded with 4% nitrate alcohol for about 10 s after grinding and polishing. The microstructure of the welded and repair-welded joints is observed by the Olympus-DSX510 optical microscope. The tensile specimens were produced in compliance with ISO 6892-1: 2009 [25], and the dimensions are presented in Figure 3. The 100 kN universal material testing machine was employed for the tensile test, and the rate of strain is 0.03/s. The hardness test was carried out on the ARTCAN-300SSI-C microhardness machine. The interval of adjacent test points is 0.5 mm, the load is 0.98 N, and the hold time is 15 s.

#### 2.2.2. High Cycle Fatigue Test

To explore the fatigue property of welded and repair-welded joints, the HCF test was conducted by the MTS high-frequency fatigue testing machine shown in Figure 4a. T-joints are fixed by the hydraulic wedge, and clamping length is 80 mm, see Figure 4b.

Table 3 lists the parameters of the HCF test. The welded joints are denoted and referred to in this study as A, while the toe repair-welded joints and deck repair-welded joints are marked as B-DZ and B-D, respectively. Each group contains four specimens, which are subjected to fatigue loads of 36/360 kN, 40/400 kN, 44/440 kN and 46/460 kN, respectively.

### 2.3. Traction Structural Stress Method

When both ends of the specimen are strained, the stress distribution around the weld is highly nonlinear in the direction of the thickness of the deck and difficult to be calculated directly. Dong [23] proposed that the stress in the direction of the thickness of the deck can be decomposed into two parts based on stress linearization. The first part is the sum of membrane stress $\sigma_{m}$ and bending stress $\sigma_{b}$, which is called traction structural stress $\sigma_{s}$, and it is balanced with external load [26,27]. The other part is notch stress $\sigma_{n}$ generated by local notch effect, which is highly nonlinear and in self-equilibrium state. The stress distribution diagram in the direction of the thickness of deck is shown in Figure 5. At present, the traction structural stress method has been proven to be suitable for describing the four categories of cracks on rib-deck T-joints. This method converts the joint force/bending moment obtained by finite element analysis (FEA) into membrane stress and bending stress to calculate the traction structural stress of the potential failure interface.

In this paper, the FEA based on ABAQUS 6.14 software is used to calculate the traction structural stress of the potential failure surface. To establish the finite element model (FEM) of joints accurately, WiKi-SCAN 2.0 is used to measure the dimensional parameters of joints, see Figure 6. The dimensional parameters of welded and repair-welded joints are listed in Table 4.

The FEM of welded and repair-welded joints are established based on the dimensional parameters, see Figure 7. Because the excess welds on the surface of the deck will be removed after repair welding, the FEM of deck repair-welded joints is the same as that of the welded joints. The boundary condition and loading mode of the welded and repair-weld joints model are the same. The degree of freedom in the Y direction of the fixture position of the FEM is constrained, that is, the surface area which is 80 mm away from both ends of the FEM. The movement at the left end of the FEM is restricted in the X and Z directions. At the same time, 1 MPa tensile stress is applied to the right end face of the FEM. It can be seen that the mesh division of the FEM is not uniform. This arrangement can not only improve the calculation efficiency but also ensure the accuracy of the calculation results because of the mesh insensitivity of the traction structural stress method.

Nodal forces calculation at the potential failure surface is shown in Figure 8. The surface of interest is defined in FEM, namely toe failure surface (Figure 8a) and root failure surface (Figure 8b). All elements connected to one side of the potential failure surface are identified to extract nodal forces, which are highlighted in Figure 8. Then the traction structural stress $\sigma_{s}$ can be calculated using Equations (1)–(3).
(1)$$\sigma_{m}=\frac{1}{t}\sum_{i}^{N}F_{i}$$
(2)$$\sigma_{b}=\frac{1}{t^{2}}\sum_{i}^{N}F_{i}\left(y_{i}-\frac{t}{2}\right)$$
(3)$$\sigma_{s}=\sigma_{b}+\sigma_{m}$$

Once the traction structural stresses have been determined using the procedures described through the above process, the equivalent traction structural stress commonly used to evaluate the fatigue life of welded joints in engineering can be obtained through Equations (4)–(7) [26].
(4)$$\Delta S_{S}=\frac{\Delta \sigma_{s}}{t^{\left(2-m\right)/\left(2m\right)}I{\left(r\right)}^{1/m}}$$
where:(5)$$I{\left(r\right)}^{\frac{1}{m}}=0.0011r^{6}+0.0767r^{5}-0.0988r^{4}+0.0946r^{3}+0.0221r^{2}+0.014r+1.2223$$
(6)m=3.6
(7)$$r=\frac{\left|\sigma_{b}\right|}{\left|\sigma_{b}\right|+\left|\sigma_{m}\right|}$$

## 3. Microstructure and Mechanical Properties

### 3.1. Microstructure

Figure 9 illustrates the variation in the microstructure of the S355J2 steel-welded joint. It can be seen that the welded joint of fusion welding is mainly composed of weld metal (WM), heat-affected zone (HAZ), and base metal. HAZ can be divided into three distinct zones, namely, coarse grain HAZ (CGHAZ), fine grain HAZ (FGHAZ), and inter-critical region (ICHAZ-lying between the base metal and FGHAZ) [28]. The WM (Figure 9a) is mainly composed of the pro-eutectoid ferrite precipitated along the grain boundary of the columnar crystal and the inner acicular ferrite. The CGHAZ (Figure 9b) consists of lath martensite and a little granular bainite. The grains continue to grow after reaching austenitizing temperature, and the overheated structure with coarse grains is obtained. The microstructure of FGHAZ (Figure 9c) contains ferrite, pearlite, and a small amount of granular bainite. Compared with the base metal, the grain size of FGHAZ is small. In ICHAZ (Figure 9d), pearlite is first austenitized and then forms finer ferrite and pearlite after cooling. Ferrite fails to austenitize but continues to grow. Therefore, the microstructure structure of ICHAZ is similar to that of the base metal, and the size of grains varies in this zone.

Figure 10 shows the microstructure of the toe repair-welded joint. The grain distribution and morphology in the repair-welded joint are similar to that in the welded joint, which can be divided into WM, HAZ, and base metal.

The Image Pro Plus 6.0 software is used to analyze the microstructure of the welded and repair-welded joints. Figure 11 compares the image of the grains in the CGHAZ, and it is found that the austenite grains further grow. The content of lamellar martensite decreases, and the content of lath martensite increases. The average austenite grain area of the weld joint is calculated to be 1645.23 μm^2^, and that of the repair-welded joint is 3368.39 μm^2^, increased by 2.05 times. The coarse grains can decrease the mechanical properties of the welded joints, which may become a weak point in the HCF test.

### 3.2. Microhardness

To eliminate the influence of point position on hardness, it must be ensured that the test position is the same in welded and the repair-welded joints, see Figure 12a. The hardness test results are demonstrated in Figure 12b. The zone ① to the left of the black line in Figure 12b is the WM of welded joints. Due to the grain growth, the average hardness of this zone decreases by about 30 HV after repair welding. The zone ② to the right of the black line in Figure 12b is the WM of repair-welded joints, and it can be found that the hardness of this zone increases by about 60 HV. This is because repair welding remelts this zone and reduces the grain size, which results in the enhancement of hardness.

### 3.3. Tensile

The tensile properties of welded and repair-welded joints are presented in Figure 13 and Table 5. The ultimate tensile and yield strength of the welded joint and the base metal are similar. While the ultimate tensile and yield strength of the repair-welded joint decrease by 21 MPa and 40 MPa, respectively. Compared with the base metal, the elongation of the welded and repair-welded joints decreases by about 25%. This may be attributed to the fact that the coarse grains in HAZ under the weld thermal cycles reduce the toughness of the materials.

## 4. HCF Behavior of T-Joints

### 4.1. Fatigue Test Results

The HCF test results of the welded and repair-welded joints are presented in Table 6. By calculation, the fatigue properties of toe and deck repair-welded joints are reduced by about 60% and 30% under the same fatigue load compared with the welded joints.

The fracture position of each welded joint is divided into two places: weld toe and weld root. Figure 14 show pictures of A1 (fracture at weld root) and A2 (fracture at weld toe), respectively. The A-2,3,4 fractures at the weld toe and A-1 fractures at the weld root. Among four deck repair-welded joints (B-D-1,2,3,4), two breaks at the weld toe and two at the weld root. This indicates that the fatigue property of the weld toe and weld root is comparable. While toe repair-welded joints (B-DZ-1,2,3,4) are all broken at the weld root. The data in Table 4 show that the angle between the weld and deck (toe angle2) increases from 120° to 155°, reducing the stress concentration at the weld toe, which results in the failure positions of toe repair-welded joints all located at the weld root. More fracture picture of joints are show in Appendix A.

### 4.2. Fracture Characteristics

The macro-fracture characteristics of B-DZ-1 (fracture at weld root) and B-D-1 (fracture at weld toe) are presented in Figure 15. It can be seen that the HCF fracture surfaces are divided into fatigue crack initiation, propagation, and fast fracture zone.

The fatigue crack of B-DZ-1 initiates on the surface of the specimen and propagates along the direction of the thickness of the deck. The color of the crack initiation is dark. After the fatigue crack propagation, the effective bearing area continues to decrease until the instant break. The fatigue crack of B-D-1 originates at the middle of the toe and diverges to both sides. Due to obvious welding defects such as pits and undercuts at the weld toe, it is easy to produce stress concentration and cause fatigue crack initiation.

### 4.3. Fatigue Life Assessment Based on Equivalent Traction Structural Stress

Through calculation of the procedures in Section 2.3, the equivalent traction structural stress of the toe surface and root surface of the welded joint is 1.599 MPa and 1.687 MPa, respectively. The equivalent traction structural stress of the toe surface and root surface of the toe repair-welded joint is 1.608 MPa and 1.679 MPa, respectively. The equivalent traction structural stress under other fatigue loads can be calculated based on these data, as demonstrated in Table 7.

In engineering, Equation (8) is used to evaluate the fatigue life of welded structures through equivalent traction structural stress.
(8)$$N={\left(\Delta S_{S}/C_{d}\right)}^{-1/h}$$

The $C_{d}$ and h are constant parameters obtained by fitting a large number of fatigue test data. Table 8 lists the constant parameters of the master S-N curve provided by ASME standard [29].

In the actual welding process, the AM generated by nonuniform temperature field distribution, assembly clamping, and other factors will affect the welding joint’s fatigue performance [30]. However, the equivalent traction structural stress calculated above does not consider the influence of AM, so the introduction of a correction formula can further improve the accuracy of fatigue life evaluation of the welded joints. Figure 16 is the schematic diagram of the AM of the welded joint. The welding root is set as the critical position, and t (mm) is the path length of fatigue crack growth, namely, the thickness of the deck plate. The *δ* (mm) is the distance between the lower edge of the misalignment side and the clamp block. The *α* (°) is the degree of AM of the specimen. The *L* (mm) is the distance between the test machine blocks. The distance between the critical position and the two blocks is defined as *L*_1_ (mm) and *L*_c_ (mm), respectively.

The position of the toe repair welding is on the same side of the deck as the original weld, so the AM is increased relative to the welded joint, see Figure 17. While the position of deck repair welding is on the different sides of the deck from the original weld, so the AM is reduced relative to the welded joint.

The AM correction coefficient $k_{\alpha }$ can be calculated by Equation (9) [31]. Then, the traction structural stress $\sigma_{s}$ is modified with AM by Equation (10).
(9)$$k_{\alpha }=\left(\frac{0.8\left(L^{4}-9L^{3}L_{c}+39L^{2}L_{c}{}^{2}-60L_{c}{}^{3}L+30L_{c}{}^{4}\right)}{L^{3}t}\right)\alpha$$
(10)$$\sigma_{sm}=\sigma_{s}\left(1+k_{\alpha }\right)$$

The AM *α* of each specimen can be obtained by using three-dimensional scanning technology and computer-aided design (CAD). The detailed parameters of the AM size of each specimen and the calculation of the correction coefficient $k_{\alpha }$ are shown in Table 9.

The fatigue data with and without AM are plotted in the master S-N curve, as shown in Figure 18. The AM means that the AM is considered. It is observed that all fatigue data are within the ±95% confidence interval of the main curve. It can be observed that the data considering AM are more aggregated to the main curve, meaning that the traction structural stress method with AM is applicable to predict the fatigue life of repair-welded joints. In addition, the fatigue properties of deck repair-welded joints are higher than that of the toe repair-welded joints. Considering that the AM of toe repair-welded joints is larger than that of deck repair-welded joints, it can be determined that the larger AM will reduce the fatigue life of the repair-welded joints.

## 5. Conclusions

In this paper, the microstructure and mechanical properties of the S355J2 steel welded and repair-welded joints are studied. The HCF properties of the T-joints are evaluated by testing and FEA. Moreover, the influence of AM on HCF performance is emphatically considered. The main conclusions are as follows:Repair welding increases the grain size of WM and HAZ, which reduces the hardness of WM by about 30 HV. The tensile strength of the welded joint is the same as that of the base metal, while the tensile strength of the repair-welded joint is reduced by about 20 MPa. Compared with the base metal, the elongation of the welded and repair-welded joint decreases by about 25%.The fracture positions of the welded joints are both weld toe and root, and the ratio is even, which is the same for deck repair-welded joints cracking. However, all of toe repair-welded joints fracture at the weld root. The macro-fracture characteristics of the root surface show that the fatigue cracks initiate on the surface of the specimen and propagate along the thickness of the deck, while the cracks of the toe surface initiate at the middle of the weld toe and propagate to both sides.The fatigue data with and without AM are all within the ±95% confidence interval of the master S-N curve, while the data considering AM is less discrete to the main curve.

## Figures and Tables

**Figure 1 materials-16-03682-f001:**
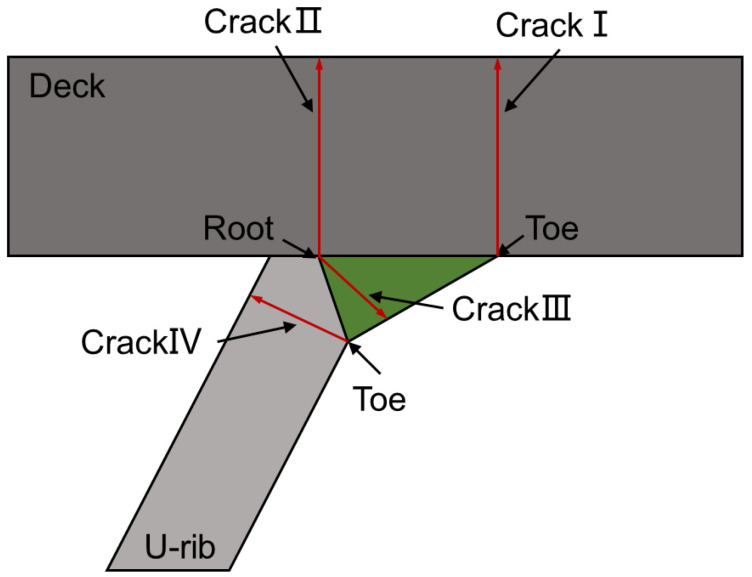
Types of orthotropic deck fatigue crack.

**Figure 2 materials-16-03682-f002:**
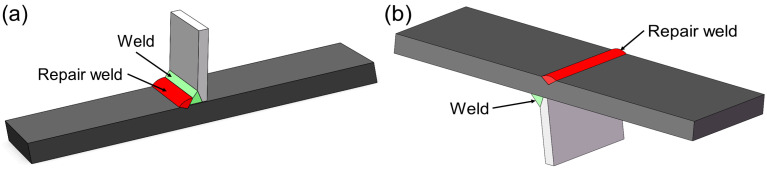
Schematic diagram of repair-welded joint: (**a**) toe repair-welded joint; (**b**) deck repair-welded joint.

**Figure 3 materials-16-03682-f003:**
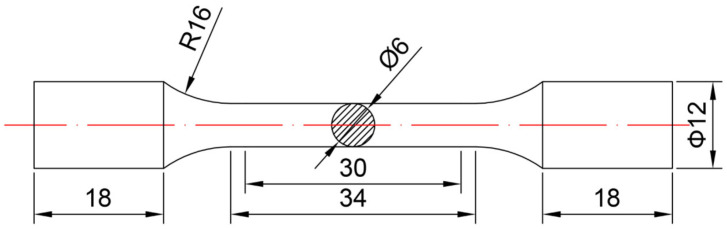
Dimensions of tensile specimen.

**Figure 4 materials-16-03682-f004:**
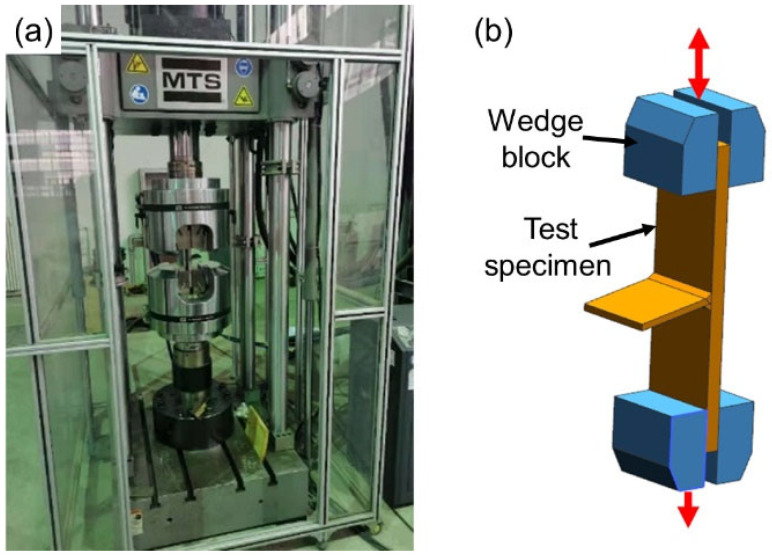
(**a**) MTS high-frequency fatigue testing machine; (**b**) clamping mode of T-joint.

**Figure 5 materials-16-03682-f005:**
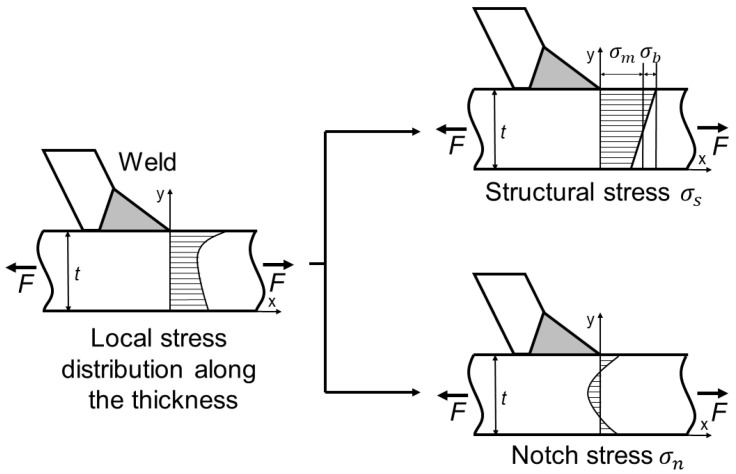
Diagram of stress distribution along the direction of the thickness of the deck.

**Figure 6 materials-16-03682-f006:**
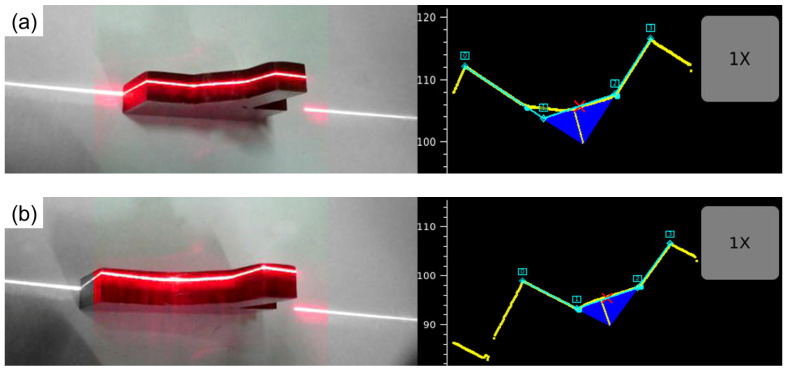
Dimensional parameters measurement of joints: (**a**) as-welded; (**b**) toe repair-welded.

**Figure 7 materials-16-03682-f007:**
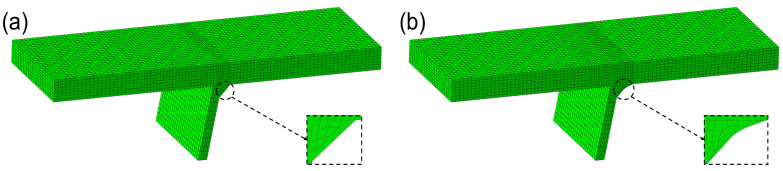
Comparison of FEM between the welded joint and repair-welded joint: (**a**) as-welded; (**b**) repair-welded.

**Figure 8 materials-16-03682-f008:**
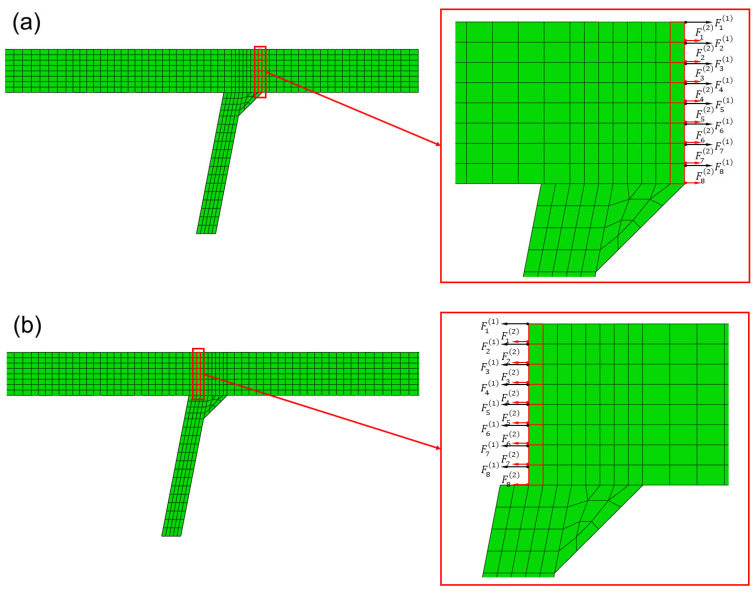
Nodal forces extraction along the weld line at weld toe and root in FEM: (**a**) weld toe; (**b**) weld root.

**Figure 9 materials-16-03682-f009:**
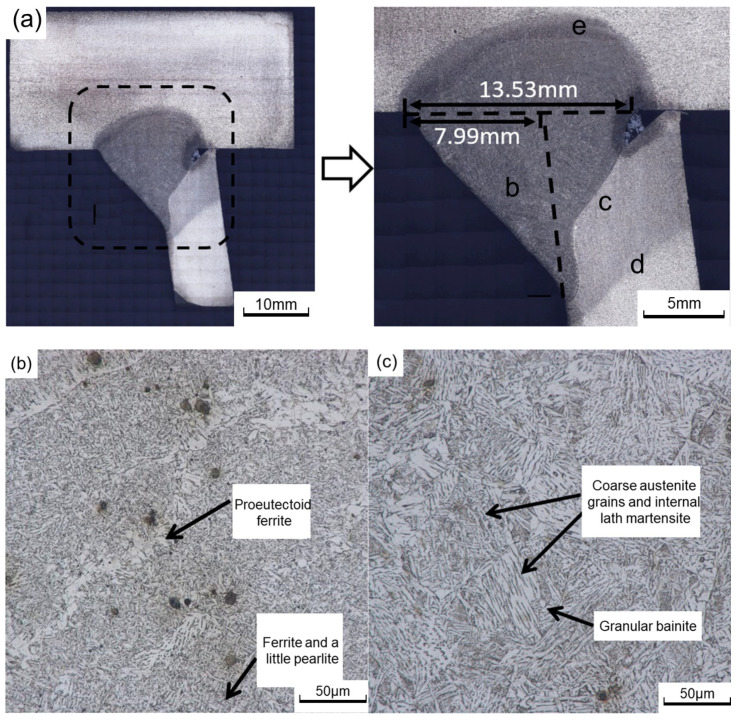
Variation of microstructure across the welded joint of S355J2 steel: (**a**) the welded joint; (**b**) WM; (**c**) CGHAZ; (**d**) FGHAZ; (**e**) ICHAZ.

**Figure 10 materials-16-03682-f010:**
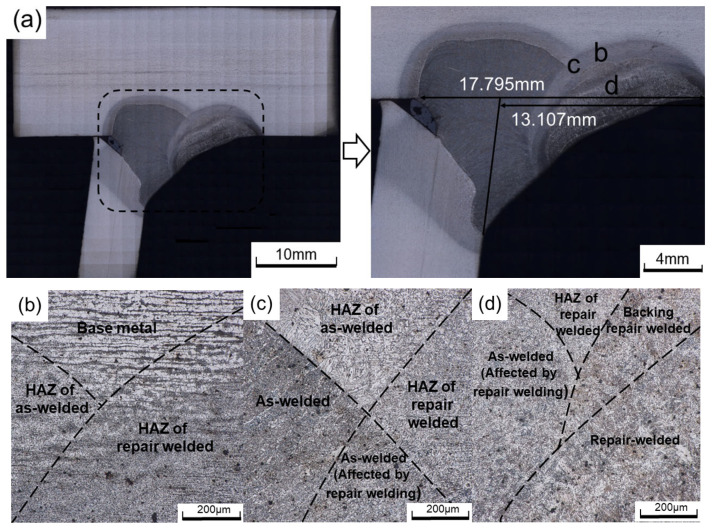
Microstructure of toe repair-welded joint: (**a**) the repair-welded joint; (**b**) region b; (**c**) region c; (**d**) region d.

**Figure 11 materials-16-03682-f011:**
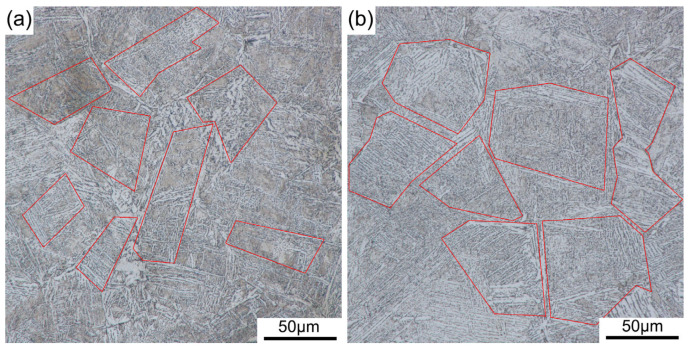
Comparison of CGHAZ microstructure of S355J2 steel joint: (**a**) as-welded; (**b**) repair-welded.

**Figure 12 materials-16-03682-f012:**
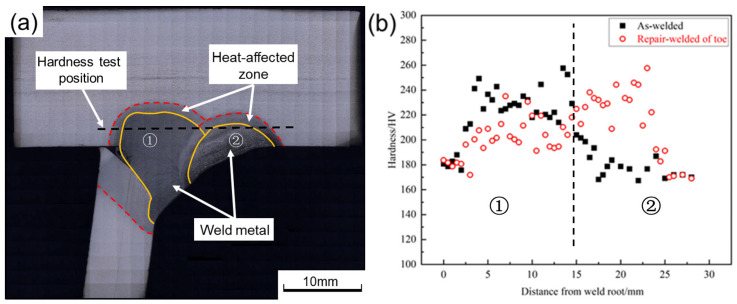
(**a**) hardness test position; (**b**) hardness test results of the welded and repair-welded joint.

**Figure 13 materials-16-03682-f013:**
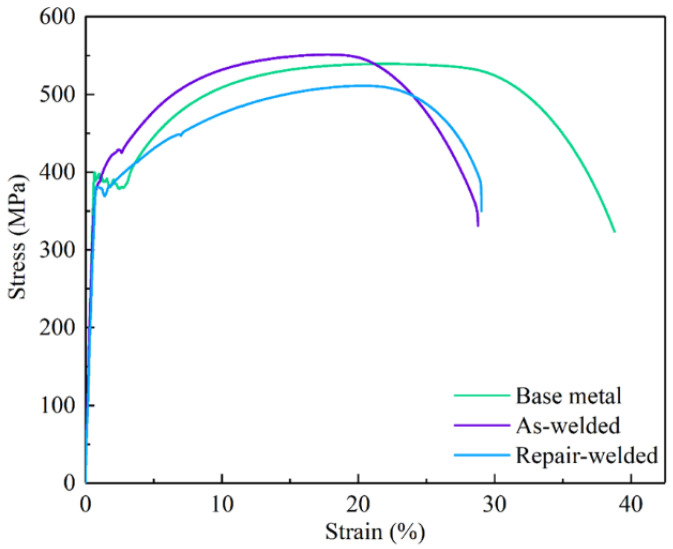
Stress-strain curves of tensile specimens.

**Figure 14 materials-16-03682-f014:**
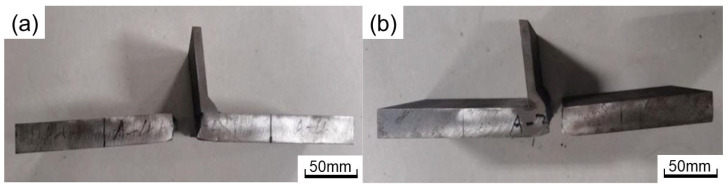
Fracture picture of welded joints: (**a**) fracture at weld root; (**b**) fracture at weld toe.

**Figure 15 materials-16-03682-f015:**
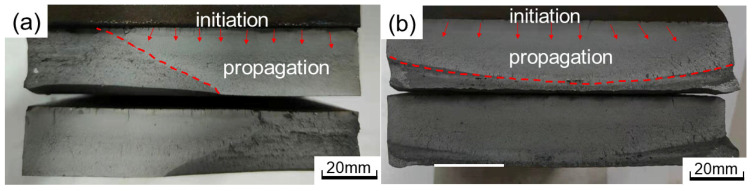
Macro-fracture characteristics of joints: (**a**) B-DZ-1; (**b**) B-D-1.

**Figure 16 materials-16-03682-f016:**
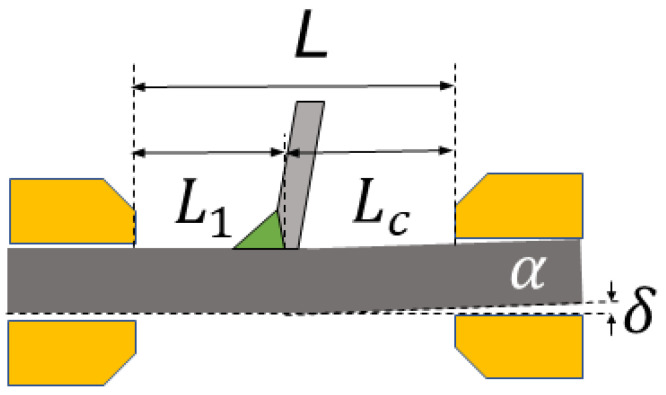
Schematic diagram of AM.

**Figure 17 materials-16-03682-f017:**
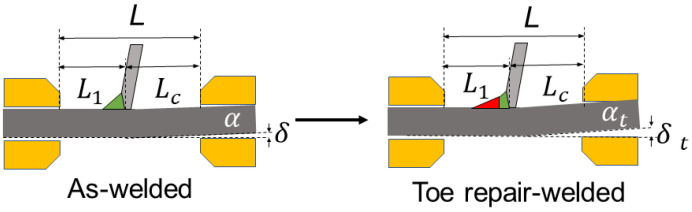
Effect of repair welding on AM.

**Figure 18 materials-16-03682-f018:**
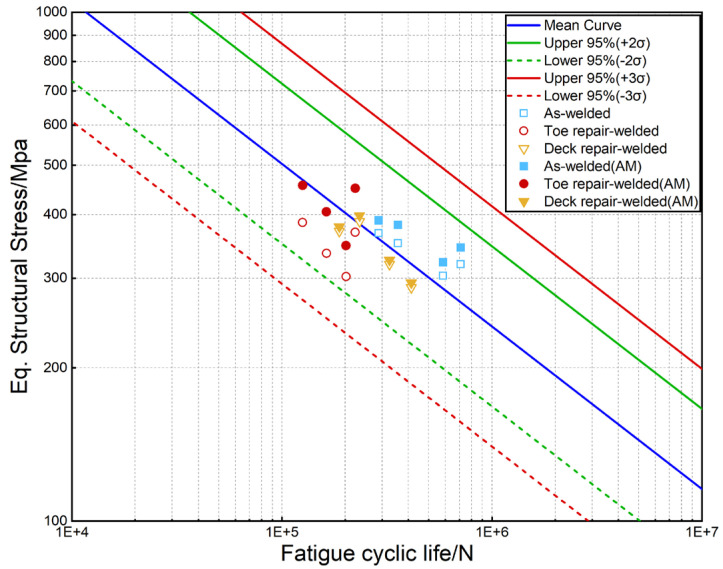
The master S-N curve of the welded and repair-welded joints.

**Table 1 materials-16-03682-t001:** Chemical composition of the base and filler metal (wt%).

Material	Si	Mn	C	P	S	Ni	Cr	Cu	Fe
S355J2	0.50	1.70	0.18	0.025	0.025	0.50	0.30	0.30	Bal.
E500T-1	0.90	1.75	0.18	0.03	0.03	-	-	-	Bal.

**Table 2 materials-16-03682-t002:** The preparation process and dimensions of the test specimen.

Specimen	Length/mm	Width/mm	Deck Thickness/mm	Schematic Diagram of Specimen
Post-weld specimen	1800	600	18	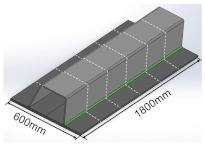
Post-cut specimen	600	300	18	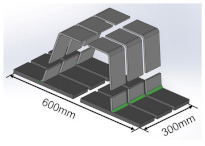
T-joint	300	100	18	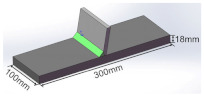

**Table 3 materials-16-03682-t003:** Parameters of the HCF test.

Specimen	Welding State	Loading Frequency/Hz	Minimum/Maximum Load/kN
A-1	As-welded	15	36/360
A-2	As-welded	15	40/400
A-3	As-welded	15	44/440
A-4	As-welded	15	46/460
B-DZ-1	Repair-welded of deck	15	36/360
B-DZ-2	Repair-welded of deck	15	40/400
B-DZ-3	Repair-welded of deck	15	44/440
B-DZ-4	Repair-welded of deck	15	46/460
B-D-1	Repair welded of toe	15	36/360
B-D-2	Repair welded of toe	15	40/400
B-D-3	Repair welded of toe	15	44/440
B-D-4	Repair welded of toe	15	46/460

**Table 4 materials-16-03682-t004:** Dimensional parameters of welded and repair-welded joints.

Specimen	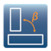 Beta Angle(°)	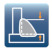 Leg1(mm)	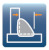 Leg2(mm)	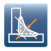 Throat(mm)	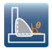 Toe Angle1(°)	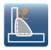 Toe Angle2(°)
Welded joints	101	10.1	7.99	5.4	146	120
Repair-welded joints	99	10.0	13.1	5.9	144	155

**Table 5 materials-16-03682-t005:** Data of the tensile tests for the different conditions.

Specimen	Yield Strength *σ_YS_*/MPa	Ultimate Tensile Strength *σ_UTS_*/MPa	Tensile Elongation *δ*/%
Base metal	393	539	38.7
As-welded	401	551	28.8
Repair-welded	380	511	29

**Table 6 materials-16-03682-t006:** HCF test results of S355J2 welded and repair-welded joints.

Specimen	Minimum/Maximum Load (kN)	Test Cycle	Fracture Location
A-1	36/360	584,810	Weld root
A-2	40/400	709,590	Weld toe
A-3	44/440	356,690	Weld toe
A-4	46/460	288,464	Weld toe
B-DZ-1	36/360	202,070	Weld root
B-DZ-2	40/400	163,023	Weld root
B-DZ-3	44/440	223,383	Weld root
B-DZ-4	46/460	125,140	Weld root
B-D-1	36/360	413,257	Weld toe
B-D-2	40/400	325,298	Weld toe
B-D-3	44/440	187,688	Weld root
B-D-4	46/460	234,109	Weld root

**Table 7 materials-16-03682-t007:** Equivalent traction structural stress of FEM under different load.

Specimen	Minimum/Maximum Load (kN)	Nominal Stress (MPa)	Stress Concentration Factor of Fracture Interface	Equivalent Traction Structural Stress of Fracture Interface (MPa)
A-1	36/360	180	1.687	303.61
A-2	40/400	200	1.599	319.80
A-3	44/440	220	1.599	351.78
A-4	46/460	230	1.599	367.77
B-DZ-1	36/360	180	1.679	302.18
B-DZ-2	40/400	200	1.679	335.76
B-DZ-3	44/440	220	1.679	369.34
B-DZ-4	46/460	230	1.679	386.12
B-D-1	36/360	180	1.599	287.82
B-D-2	40/400	200	1.599	319.80
B-D-3	44/440	220	1.687	371.10
B-D-4	46/460	230	1.687	387.95

**Table 8 materials-16-03682-t008:** Parameters of the master S-N curve.

Curve	$C_{d}$	h
Master curve	19,930.2	0.3195
Upper curve (+2σ)	28,626.5	0.3195
Lower curve (−2σ)	13,875.7	0.3195
Upper curve (+3σ)	34,308.1	0.3195
Lower curve (−3σ)	11,577.9	0.3195

**Table 9 materials-16-03682-t009:** Parameters of the AM and the correction coefficient of each specimen.

Specimen	*L* (mm)	*L*_c_ (mm)	*α* (°)	$k_{\alpha }$	$\sigma_{sm}\;(\mathrm{MPa})$
A-1	140	69.2	0.99	0.0672	322.56
A-2	140	69.2	1.23	0.0835	344.56
A-3	140	69.2	1.36	0.0923	381.90
A-4	140	69.2	0.95	0.0645	389.75
B-DZ-1	140	69.2	2.385	0.1618	347.61
B-DZ-2	140	69.2	3.273	0.2221	405.02
B-DZ-3	140	69.2	3.497	0.2373	450.74
B-DZ-4	140	69.2	2.9	0.1968	456.70
B-D-1	140	69.2	0.342	0.0232	294.00
B-D-2	140	69.2	0.303	0.0206	325.88
B-D-3	140	69.2	0.328	0.0223	378.76
B-D-4	140	69.2	0.419	0.0284	398.20

## Data Availability

Not applicable.

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
