# Peer review of "Influence of Repair Welding on the Fatigue Behavior of S355J2 T-Joints"

_materials, 2023, doi:10.3390/ma16103682_

Round 1
Reviewer 1 Report
The authors investigate the influence of repair welding of the S355J2 steel in T-joints orthotropic bridge decks. They analyze the microstructure, mechanical properties, and high-cycle fatigue properties using metallographic methods, tensile test and harness, and MTS high-frequency fatigue, respectively. The figures are very illustrative with good resolution. One of the main conclusions is related to accurately predicting fatigue life using the equivalent traction structural, which agrees with previous studies. The authors also use FEM simulation, but it only appears in the result section and is presented as a mixture between methods, modeling, and results, which is difficult to follow. I recommend including section 2.3. Finite Element method (FEM) in the methodology and describe modeling and methods adopted in the FEM. Also, Create the results section, which does not appear during the paper. In general, the paper has content to be published, it just needs to improve its presentation.
Reviewer 2 Report
Notes are attached below

Notes are attached below
Reviewer 3 Report
In this paper, the authors tried to investigate microstructure, mechanical properties and high-cycle fatigue behavior of T-joint welds in two states of initial and repair welding. To this end, many experiments have been conducted and valuable results have been reported. They also tried to provide a finite element model for it and numerically achieve similar results. They were successful. In addition, the article is well written and well structured. Also, the authors described the events in an eloquent and understandable language for everyone. The most important advantage of this article is the detailed interpretation of the obtained results and their discussion, which shows the expertise of the authors in this field. Moreover, very good schematic images are used to explain the details. In summary, I believe that this article can be published in the journal in the current format. However, here are a few points for minor revision:
1- Please add an appropriate reference to Table 1.
2- Has the reproducibility of the results been investigated in the fatigue test? In other words, each sample is tested only once or tested several times at a certain stress level and the average result is reported? As you know, the phenomenon of fatigue has a lot of dispersion and it is not very pleasant to be sure of the test results based only on one sample.
3- Figures 10-12 can be located in the appendix.
4- It seems that the authors did not simulate the welding process, is it true?
